# Zinc Finger Proteins: Functions and Mechanisms in Colon Cancer

**DOI:** 10.3390/cancers14215242

**Published:** 2022-10-26

**Authors:** Shujie Liu, Xiaonan Sima, Xingzhu Liu, Hongping Chen

**Affiliations:** 1Department of Histology and Embryology, Medical College, Nanchang University, Nanchang 330006, China; 2Medical Department, Queen Mary School, Nanchang University, Nanchang 330006, China

**Keywords:** zinc finger proteins (ZFPs), colon cancer, proliferation, biological function, anticancer therapy

## Abstract

**Simple Summary:**

Over the past few decades, despite advances in colon cancer surgery, the pro-gnosis of late colon cancer patients with liver metastasis remains poor. Currently, its incidence ranks second in men and third in women. Fortunately, accumulating evidence has unraveled that zinc finger proteins (ZFPs) will shed light on the treatment of colon cancer. As the largest transcription family in the human genome, ZFPs are a class of transcription factors that can bind Zn^2+^, self-fold to form “finger” domains and regulate the expression of target genes. In this article, we elucidate the specific molecular mechanisms of ZFPs that are well-suited to orchestrate pathophysiological changes in colon cancer development, which may lay a credible foundation for further precision oncology.

**Abstract:**

According to the global cancer burden data for 2020 issued by the World Health Organization (WHO), colorectal cancer has risen to be the third-most frequent cancer globally after breast and lung cancer. Despite advances in surgical treatment and chemoradiotherapy for colon cancer, individuals with extensive liver metastases still have depressing prognoses. Numerous studies suggest ZFPs are crucial to the development of colon cancer. The ZFP family is encoded by more than 2% of the human genome sequence and is the largest transcriptional family, all with finger-like structural domains that could combine with Zn^2+^. In this review, we summarize the functions, molecular mechanisms and recent advances of ZFPs in colon cancer. We also discuss how these proteins control the development and progression of colon cancer by regulating cell proliferation, EMT, invasion and metastasis, inflammation, apoptosis, the cell cycle, drug resistance, cancer stem cells and DNA methylation. Additionally, several investigations have demonstrated that Myeloid zinc finger 1 (MZF1) has dual functions in colon cancer, which may both promote cancer proliferation and inhibit cancer progression through apoptosis. Generally, a comprehensive understanding of the action mechanisms of ZFPs in colon cancer will not only shed light on the discovery of new diagnostic and prognosis indicators but will also facilitate the design of novel targeted therapies.

## 1. Introduction

As one of the most common gastrointestinal malignancies worldwide, colon cancer is a tumor derived from colonic mucosal epithelium or glands. It is a widely heterogeneous disease, closely related to environmental and genetic factors [1]. Around the world, both its incidence and fatality rates have continued to climb, with approximately 1,096,000 new diagnosed cases of colon cancer in 2018, according to the GLOBOCAN 2018 database published by the International Center for Research on Cancer (IARC) [2].

At present, the treatment methods of colon cancer are still mainly surgical treatment and chemoradiotherapy, but their therapeutic effects are not ideal enough, which is mainly related to the pathological stage and invasiveness of colon cancer. For stage I colon cancer confined to the mucosal layer, the treatment is generally effective, with a patient 5-year survival rate approaching 90%, but there is a 1.2–4.9% risk of recurrence. For stage II colon cancer that is located in the colon wall but has not spread to the nearby lymph nodes, the 5-year survival rate is 65–87% [3]. For stage III colon cancer with locally advanced disease that has spread to the nearby lymph nodes but no distant metastasis, the 5-year survival rate is 53–90% after positive and comprehensive treatment, such as surgery and chemotherapy [3]. However, for stage IV colon cancer with local recurrence or distant metastasis, the prognosis is poor, and the 5-year survival rate of patients can rapidly decline to 12% [3]. Given the poor prognosis of advanced colon cancer, it is necessary to explore the molecular mechanisms underlying colorectal cancer initiation, progression and metastasis; discover the potential biomarkers associated with colorectal cancer and develop more effective therapies. In recent years, the association of proteins with colon cancer has attracted much attention from researchers, especially zinc finger proteins (ZFPs).

As an essential class of transcription factors in the human body, ZFPs have been identified to play crucial roles in several physiological processes through different molecular mechanisms, such as keratinocyte differentiation, muscle differentiation and the regulation of cancer stem cells [4]. Meanwhile, the aberrant expression of ZFPs provokes different pathological processes, such as tumorigenesis and progression, diabetes, skin diseases and neurodegeneration [5,6,7]. Especially in tumors, ZFPs have significant effects on cell proliferation, EMT, invasion and metastasis, inflammation, apoptosis, the cell cycle, drug resistance, cancer stem cells and DNA methylation in a broad range of cancers, such as colon, breast, lung and gastric cancers, as well as hepatocellular carcinoma [8] (Figure 1). In this article, we review the association of ZFPs in colon cancer. Hopefully, our review will provide directions for the targeted treatment of colon cancer, as well as exploring the possibility of using ZFPs to build a prognostic model for colon cancer.

## 2. Zinc Finger Proteins

By identifying certain DNA sequences, transcription factors play a role in the regulation of numerous biological processes, including cell differentiation, immunity, apoptosis, proliferation, autophagy and stem cell maintenance [9]. ZFPs are a superfamily of transcription factors that comprise at least one zinc finger domain that could bind specific DNA sequences, hence regulating the DNA expression levels [10]. Moreover, ZFPs can also interact with RNA, lipids, membranes and proteins through zinc finger domain or certain structural domains, such as the SCAN domain [11,12,13]. The zinc finger motif, composed of 30 cysteines and/or histidines, was firstly observed in the African Xenopus oocyte transcription factor IIIA in 1988, named for its capacity to bind zinc ions and create finger-like structures [14]. Subsequent research has revealed a significant number of proteins with zinc finger domains, which are acknowledged as the biggest family of specific transcription factors encoded by approximately 2% of human genes [15]. The zinc finger domain maintains the function and stability of the protein structure through autonomous folding upon engagement with one or more zinc ions [16]. 

According to the HUGO Gene Nomenclature Committee, the zinc finger protein family is divided into 30 types [17]. Zinc finger motifs are categorized into eight subclasses based on the structural particularity adjacent the zinc binding site, namely TAZ2 domain-like, Cys2His2 (C2H2)-like, Treble clef, Gag knuckle, zinc-binding loops, Zn2/Cys6, Metallothionein, and TAZ2 domain-like. Among them, the Cys2His2 zinc finger is the most widely distributed DNA-binding domain of the human transcription factors [18]. In addition to the variability of the zinc finger structural domains, the diversity of specific structural domains is also involved in constituting the structural and functional multiplicity of ZFPs. For instance, SET domains catalyze via the methylation of proteins, and the BTB domain primarily represses transcription by binding to and attracting corepressors of target gene transcription [19,20]. In general, the structural diversity of zinc finger proteins determines their functional complexity.

## 3. ZFPs and Their Role in Colon Cancer Patient Outcomes

There are many ZFPs that are closely related to the progression and malignancy of colon cancer, being potential indicators of patient prognosis and survival. For instance, CIZ1, ZNF217, ZNF281, ZKSCAN3, ZNF692, ZNF750, and ZFP36 proteins are closely related to the tumor volume, lymph node metastasis status, and TNM stage, and their expression levels had statistically significant effects on the survival time of patients with colon adenocarcinoma. 

In colon cancer patients, many upregulated ZFPs are closely related to their poor prognosis, and studies have investigated the specific mechanisms of action among them. It was identified that the higher the stage, the higher the positive expression rate of the CIZ1 protein, suggesting that CIZ1 may promote tumor cell infiltration and metastasis [21]. Moreover, ZNF217 is capable of promoting tumor cell invasion and metastasis through inhibiting E-cadherin expression in tumor cells [22]. The low expression of ZNF281 inhibited cell proliferation, invasion, migration, and metastasis by inhibiting the Wnt/β-catenin pathway [23]. ZKSCAN3 can transcriptionally activate integrins β4 and VEGF, promoting the occurrence of CRC [24]. ZNF692 promotes cell proliferation, migration, and invasion in colon adenocarcinoma (COAD) by downregulating p27kip1 or upregulating cyclin D1, cyclin-dependent kinase 2 (CDK2), matrix metalloproteinase-9 (MMP-9), and PI3K/Akt signaling [25]. Nonetheless, several downregulated ZFPs are substantially related to a poor prognosis in patients with colon cancer. For colon cancer patients, the low expression of ZNF750 is significantly associated with a poor prognosis [26]. The loss of ZFP36 expression occurs in the early stages of tumorigenesis, where it plays a suppressive function in the regulation of multiple cellular pathways. Furthermore, downregulated ZFP36 can advance the development of colon cancer (CT26 cell line) in mice via elevating the expression of inflammatory factors IL-23, HuR [27,28], vascular endothelial growth factor (VEGF), cyclooxygenase 2 (COX-2) [29], ZEB1, Sox9, and MACC1 [30] or increasing the stability of vitamin D receptor (VDR) [31] and claudin-1 [32]. Interestingly, only obese patients were strongly associated with the overall survival (OS) of RPS27, which promotes leptin-induced CRC through activation of the c-Jun N-terminal kinase (JNK)/c-Jun pathway [33].

In addition, the liver is the main target organ for the distant metastasis of colon cancer, which is an important factor affecting its prognosis. About 50% of patients with colorectal cancer die of colorectal liver metastasis (CALM), of which 25% were accompanied by liver metastasis when diagnosed, and 25% were found after radical surgery [34]. The median survival time of patients with unresectable liver metastases was only 6.9 months [35]. Reports have shown that ZNF185 and ZEB1 expression are significantly associated with liver metastasis and are independent indicators of liver metastasis and prognosis in colon cancer patients [36,37].

## 4. Biological Functions of ZFPs in Colon Cancer

Since there is currently no systematic review in this field, highlighting the connection between ZFPs and colon cancer is an important part of the research. Accumulating evidence suggests that ZFPs play crucial roles in cell proliferation, EMT, invasion and metastasis, inflammation, apoptosis, the cell cycle, drug resistance, cancer stem cells, and DNA methylation during colon carcinogenesis and progression (Figure 1). In the following section, we will outline the roles of ZFPs in the regulation of these biological processes and emphasize the involved molecular mechanisms (Table 1).

### 4.1. ZFPs Regulate Cell Proliferation 

Normally, cell proliferation maintains normal cell numbers through a finely coordinated network of external growth factors and intracellular gene regulation [67,68]. However, uncontrolled or sustained cell proliferation is a typical hallmark of cancer, in which cancer cells can evade strict surveillance and promote cancer progression [69]. Limiting the proliferation of cancer cells is a direction for cancer therapy; therefore, it is crucial to identify more molecular targets related to cancer cell proliferation [70]. According to previous studies, multiple ZFPs have regulatory effects on cell proliferation in colon cancer.

Several ZFPs have been discovered to be upregulated in colon cancer and have a proliferative effect. ZNF398 (Krüppel C2H2 type) is a heterodimer from the zinc finger protein ZER6, which contains a Krüppel-associated box (KRAB) domain at the N-terminal end [40]. It was found that ZNF398 was highly expressed in colon cancer, while ZNF398 knockdown resulted in constrained cell proliferation. Further studies revealed that p52-ZER6 strengthens the MDM2/p53 complex by attaching to p53 via the KRAB (tKRAB) structural domain. Then, the increased ubiquitination and degradation of p53 induces dysregulation of the cell cycle and eventually results in unchecked cell growth [71]. In addition, Ma et al. found that ZFP91 (C2H2-type) was upregulated in human colon cancer cells and positively linked with hypoxia-inducible factor-1 (HIF-1) expression [44]. As a widely recognized cancer therapeutic target, HIF-1 upregulation is known to have a significant role in cell survival, tumor metastasis, and poor prognosis [72]. In terms of the mechanism, ZFP91 functions as a driver gene to activate NF-κB/p65, which results in the upregulation of HIF-1α expression, ultimately leading to the excessive proliferation of colon cancer cells [47]. Moreover, Myeloid zinc finger 1 (MZF1) is a ZFP from the Krüppel family containing a SCAN domain and was found to be aberrantly expressed in several cancers [73]. It was found that, in colon cancer cells, MZF1 promotes proliferation by binding to the promoter of the receptor tyrosine kinase (Axl) and activating the promoter, thereby increasing the expression level of Axl [74]. Xing et al. discovered that ZNF692 (Krüppel C2H2 type) is enhanced in colon adenocarcinoma (COAD) and encourages the G1/S phase transition, which, in turn, boosts tumor cell proliferation [25]. Additionally, ZFP185 (LIM type) is regarded as an independent predictor of liver metastasis and prognosis in patients with colon cancer and has been linked to the control of cell proliferation and differentiation [75]. Furukawa et al. discovered that PATZ1 (Krüppel C2H2 type) increases colon cancer cell proliferation via activating the ERK/MAPK pathway and is increased in colon cancer cells. The scientists also speculate that PATZ1 may be a potential proto-oncogene for colon cancer, although the precise mode of action has yet to be investigated [76].

The Krüppel-like factors (KLF) family is a class of ZFPs containing 17 members, and earlier research has indicated that the KLF family is essential for the development and prognosis of a few malignancies [77]. The downregulation of KLF4 was related to a poor prognosis in renal cell carcinoma, for instance [42]. A recent study found that the increased expression of some KLF family members was associated with worse overall survival in patients with colon cancer, such as KLF17, KLF14, KLF2, and KLF15. In contrast, some ZFPs from the KLF family are aberrantly expressed in colon cancer and have proliferation inhibitory effects, such as KLF5, KLF4, KLF6, and KLF3 [43]. In detail, KLF4 was observed to be expressed at enhanced levels in the skin and intestine. Yu et al. showed that the overexpression of KLF4 suppresses colon cancer cell proliferation via directly inhibiting B lymphoma Mo MLV insertion region 1 (Bmi1), a polyclonal group (PcG) protein that participates in the regulation of colon cancer cell proliferation [41]. Additionally, KLF6-SV2, an SV2 variant of one of the KLF6 selective splice isoforms, was discovered to be considerably inhibiting the proliferation of the colon cancer (SW480 and SW620 cell lines) [78]. Overall, the role and mechanisms of other KLF family members in colorectal cancer have not been elucidated fully yet.

### 4.2. ZFPs Regulate EMT and Promote Invasion and Metastasis

A crucial step in the early stages of tumor invasion and metastasis creation is the epithelial–mesenchymal transition (EMT), a biological process in which epithelial cells lose cell polarity, gain a greater capacity for invasion and migration, and destroy the extracellular matrix. Studies have demonstrated that a substantial number of ZFPs can facilitate this process by controlling the EMT. Predominantly, EMT is recognized by the decreased expression of cell adhesion molecules such as E-cadherin. Usually, the expression of the E-cadherin (CDH1) gene is tightly regulated. Furthermore, it is revealed that the zinc finger transcription factors ZNF143, Twist, Snail family, ZEB1, and ZEB2, as molecular switches of EMT, participate in its regulation. 

In colon cancer patients, ZNF143 downregulation was observed. Further research found that ZNF143 knockdown can increase the expression of transcriptional ZEB1, thus inhibiting the transcriptional expression of E-cadherin and promoting EMT transformation [38]. However, the upregulation of many other ZFPs was found in colon cancer patients, promoting EMT, invasion, and metastasis through multiple pathways. The upregulation of Twist in colon cancer (SW480, HCT116, and HT29 cell lines) not only results in the high transcriptional expression of vimentin, N-cadherin, and fibronectin but also decreases the expression of E-cadherin, which could promote EMT and enhance the metastatic ability of tumor cells [52,79]. Additionally, it is worth mentioning that the relationship between the Snail family and E-cadherin in colon cancer is a certain controversy. Snail is a major inducer of the EMT process in colon cancer. It can bind to the E-box DNA sequence of the E-cadherin promoter and recruit G9a [58], lysi-specific demethylase 1 (LSD1) [59], histone deacetylases (HDACs) [57], Polycomb inhibitory complex 2 (PRC2) [80,81], and hybrid 3–9 homolog 1 (SUV39H1) inhibitors, resulting in various histone modifications and inhibiting E-cadherin transcription. However, Kroepil F et al. showed that, in whole colon tumors, there was no significant correlation between Snail and E-cadherin expression and E-cadherin deletion or even downregulation [82]. In addition, the ZEB family of the zinc finger transcription factors is also one of the main inducers of EMT, especially ZEB1 and ZEB2, which can directly or indirectly limit the expression of adhesion proteins such as E-cadherin and a series of polar proteins at the transcriptional level. They mainly mediate transcriptional repression by raising the corepressor C-terminal-binding protein (CTBP) to the E-box [83]. Meanwhile, they could also inhibit the transcription of E-cadherin and induce EMT through other multiple pathways. For instance, they are capable of recruiting the switch/sucrose nonfermentable (SWI/SNF) chromatin remodeling protein BRG [54], interacting with the transcriptional coactivator p300/CBP-related factor (PCAF) to promote Smad signal transduction [84], or forming a protein complex with human telomerase reverse transcriptase (hTERT) to bind to the E2 box conserved in the promoter region of the CDH1 gene [55]. Moreover, ZEB1 can promote colon cancer invasion by regulating the molecules involved in matrix remodeling, such as uPA and plasminogen activator inhibitor-1 (PAI-1) [32]. Intriguingly, the ZEB family is regulated by upstream Snail factors at the transcriptional and post-transcriptional levels; meanwhile, Twist can directly enhance the Snail expression [60].

### 4.3. ZFPs Regulate Inflammation 

In recent years, studies have shown that inflammatory bowel disease (IBD) is one of the important mechanisms and high-risk factors for the development and progression of colorectal cancer (CRC) [85]. The underlying mechanism may provide cancer cells, the surrounding cellular stroma, and inflammatory cells to participate in a well-orchestrated inflammatory tumor microenvironment (TME). Multiple studies have shown that ZFPs, such as ZNF281, ZFP91, ZNF70, and MAZ, can affect the immune response process and promote cancer progression. 

In colon cancer patients, serval ZFPs are upregulated and contribute to the development of disease through inflammatory physiological processes. High expression of ZNF281 could increase the expression of inflammatory cytokines (IL-8, IL-1β, IL-17, and IL-23) gene expression, which promotes the inflammation-induced elevation of extracellular collagen levels and morphological changes [46]. What is more, ZFP91 can not only activate HIF-1α through NF-κB/p65 [44] but also positively regulate the production of inflammatory cytokine IL-1β in macrophages by activating the MAPKs and atypical caspase-8 inflammasome [45,48], which advances the proliferation and tumorigenesis of colon cancer. Previous studies have illustrated that ZNF70, as a target between macrophages and colon cancer cells, promotes the secretion of IL-1β by macrophages by regulating the activation of NLRP3 inflammasome and STAT3 in macrophages (THP-1) and then boosts the proliferation of colorectal cancer cells (HCT116 cell line) [64]. In addition, the AAV-mediated silencing of ZNF70 in the AOM/DSS model inhibited IL-1β secretion in the mouse serum and tumor growth in the CRC model of colitis. In addition, MAZ plays a crucial role in the transcription of inflammatory target genes, such as tumor necrosis factor α (TNF-α) and neutrophil chemokine Cxcl1, directly through activating the hypoxia-induced transcription factor (HIF-2a) in the progression of colitis hypoxia [49,86]. In turn, STAT3 phosphorylation is dependent on HIF-2A-independent activation. Furthermore, abnormalities in the signaling pathway with STAT3 have been found in both ulcerative colitis and Crohn’s disease, suggesting an important role in the progression of ulcerative colitis [87]. Evidence provides a certain clue and theoretical basis for the development of inflammation-related colon cancer-targeted drugs with the advantages of strong targeting, fine mapping, and few side effects.

### 4.4. ZFPs Regulate Apoptosis 

Apoptosis is an evolutionarily conserved programmed cell death process that does not cause inflammatory responses, which is essential for animal development and tissue homeostasis. Researchers have demonstrated that some ZFPs are vital in colon cancer progressions, such as ZNF545, KLF6-SV2, ZEB2-AS1, BORIS, and ZNF750. 

ZNF545 and KLF6-SV2 were significantly downregulated in colon cancer patients. As a novel tumor suppressor, ZNF545 can induce tumor cell apoptosis and inhibit ribosomal protein translation and target gene transcription [88]. Wang et al. found that the overexpression of ZNF545 in CRC cells induced growth arrest and apoptosis. The underlying mechanism may be that ZNF545 bind to the minimal rDNA promoter by its two zinc finger clusters, where it interacts with KAP1 to assemble a transcriptional repressor complex [66]. Moreover, KLF6-SV2 can also act as a tumor suppressor by effectively blocking colorectal cancer cell proliferation, arresting the cell cycle, and inducing apoptosis, which may be associated with the enhanced expression of p21 and Bax [78]. Additionally, several ZFPs are significantly upregulated in colon cancer patients. In colon cancer cells (DLD1 and SW620), ZEB2-AS1 not only induced β-catenin expression, the activation of downstream target gene transcription, promotion of proliferation, migration, and invasive ability but also inhibited apoptosis [89]. Furthermore, BORIS, a novel oncogene, is an 11-zinc finger (ZF) protein [61]. In the physiological state, it is not found or detected at very low levels in human tissues or cells, but it is activated and expressed in cancer due to hypomethylation of its promoter. The silencing of BORIS can induce apoptosis in tumor cell lines. Nguyen, P. et al. discovered that BORIS readily binds to unmethylated DNA-binding sites but preferentially binds to paternally H19 differentially methylated regions [90]. Sun, L. et al.’s experiments showed that this may be due to BORIS acting together with DNA methyltransferases 1 and 3b (DNMT1 and dnmt3d) to regulate promoter histone methylation and activate the expression of the multifunctional protein BAG-1, which can interact with several target molecules, thus regulate apoptosis, proliferation, transcription, metastasis, and motility [91]. In addition, Xia et al. proved that a high expression of ZNF750 promoted proliferation, motility, and invasion; inhibited apoptosis in human colon cancer cell lines (SW620, HCT116, Caco2, and SW480); and the underlying mechanism was that ZNF750 could positively regulate the expression of long noncoding RNA CYTOR, enhance the tumorigenicity of colon cancer cells, and affect the response to tumor drug treatment [92].

### 4.5. ZFPs Regulate Cell Cycle 

The strict control of the cell cycle is required for normal cell proliferation, in which cells undergo the doubling of chromosomes and other cellular material before dividing into two daughter cells surrounded by membranes, a process also known as mitosis [93]. Under regular conditions, the cell cycle is tightly regulated by CDKs (cell cycle protein-dependent kinases) and cell cycle proteins, and CKIs (cell cycle protein-dependent kinase inhibitors) inhibit the actions of CDKs. In addition, the tumor suppressor genes p53 and RB are also involved in the control of the cell cycle [94]. In cancer cells, cell cycle dysregulation induces uncontrolled proliferation, which promotes tumor development and metastasis [95]. Several studies have shown that numerous ZFPs are involved in the regulation of the cell cycle in colon cancer (Figure 2).

KLF6-SV2, also known as ZFP9, was discovered to be expressed at decreased levels in colon cancer. In the KLF6-SV2 overexpression model constructed by the researchers, a suppressed cell cycle and cell proliferation were observed. Additionally, it was shown that KLF6-SV2 may limit the cell cycle of cancer cells by enhancing the expression levels of Bax and p21 [78]. In colon cancer tissues, ZNF398 was overexpressed, and the silencing of ZNF398 resulted in G0–G1 phase obstruction. Huang et al. found that ZNF398 promotes p53 ubiquitination through the binding of the KRAB structural domain with p53, which reinforces the MDM2/p53 complex [71]. Furthermore, it was demonstrated that ZFP91 promotes colon cancer cell proliferation by activating HIF-1α via NF-κB/p65 and leading to a significant increase in the proportion of cells in the S phase [44]. By triggering the phosphatidylinositol 3-kinase (PI3K)/AKT serine/threonine kinase (AKT) signaling cascade, ZNF692 (Krüppel C2H2 type) overexpression promotes the G1/S phase transition while controlling the production of p27Kip1, cyclin D1, matrix metalloproteinase-9 (MMP-9), and cyclin-dependent kinase 2 (CDK2) [25]. PATZ1 knockdown dramatically triggered cell cycle arrest by lowering cell cycle protein D1/E1 and raising the p21 expression levels, in addition to its involvement in supporting colon cancer cell proliferation [50]. Additionally, Kim et al. demonstrated that Slug increased p53 ubiquitination/degradation in colon cancer cells and suppressed the function of p53 and p21 by increasing the expression level of Mdm2 [53].

### 4.6. ZFPs Regulate Drug Resistance 

As a dilemma in colon cancer treatment, most patients experience drug resistance and result in poor treatment outcomes. There are multiple processes that lead to the development of drug resistance and are classified as cellular and noncellular [96]. Identifying the molecules involved in the contribution to drug resistance would assist in reversing drug resistance and improving the treatment outcomes. Interestingly, studies have shown that zinc finger proteins are involved in the development of drug resistance in colon cancer.

In the research carried out by Deng et al., the result of the MTT assay showed that human colon cancer cell lines (SW480 and HCT116) with upregulated Twist levels showed a decreased sensitivity to oxaliplatin. Additionally, the overexpression of Twist also induced a higher mRNA level of ABCB1, which is a chemoresistance-related gene, and encoded transmembrane protein P-gp (P-glycoprotein) [51]. Furthermore, it is interesting to note that drug-resistant lung cancer cells were observed to exhibit cancer stem-like traits and an EMT phenotype, indicating that drug resistance caused by Twist in colon cancer may be linked to its role in cancer stem cells and EMT. Further research is needed to determine the exact mechanism, though [97].

### 4.7. ZFPs Regulate Cancer Stem Cell 

Cancer stem cells (CSCs) are a subpopulation of tumor cells that undergo Hedgehog and Notch signaling and are responsible for tumor self-renewal, maintenance, and growth. Several studies have revealed that tumor stem cells aid in the spread of cancer and increase its resistance to standard cancer treatments such as chemotherapy and radiation. Accordingly, CSCs hold great potential as therapeutic targets [98]. ZFX is a zinc finger transcription factor implicated in the malignant development of hepatocellular carcinoma by modulating the expression of CSC markers such as Nanog and SOX-2 [62]. Additionally, ZFX has been closely linked to the prognosis of patients with colorectal cancer, and Yan et al. hypothesized that ZFX may enhance CRC cells via generating stem cells. However, the exact mechanism remains more investigated [63]. In addition to increasing metastasis, there is evidence that a high expression of Twist in colon cancer cells confers tumor stem cell-like features to cancer cells [51]. These ZFPs involved in the control of tumor stem cells are anticipated to be potential colon cancer therapeutic targets.

### 4.8. ZFPs Regulate DNA Methylation 

DNA methylation, one of the epigenetic processes that regulate gene expression, modifies the genetic expression without affecting the DNA sequence. Significantly, the development of cancer is accompanied by substantial intracellular DNA methylation alterations, and DNA methylation in cancer has various roles that contribute to the transformation of healthy gene expression regulation into a disease pattern [99]. A rising corpus of research demonstrates the considerable potential of DNA methylation alterations for the development of cancer diagnostic tests with significant therapeutic relevance [100]. Lindner et al. revealed that ZEB1 impacts the expression of chromatin-modifying enzymes in colon cancer. Mechanistically, ZEB1 might attract histone tyrosine kinase HDAC1 or methyltransferase DNMT1 to the E calmodulin promoter, inhibiting their transcription and preserving their hypermethylated state, respectively [56].

## 5. MZF1: A Double-Edged Sword in Colon Cancer

Typically, ZFPs tend to exert only one of these two effects, anticancer or oncogenic. However, in the past few years, there have been several studies suggesting that one ZFP may have dual roles in colon cancer (Figure 3). Myeloid zinc finger protein 1 (MZF1), which belongs to the scan-ZF family of transcription factors [101], is located in the chromosome 19q13.43 region. As a transcription factor, MZF1 binds to target DNA promoters such as padi1 and cdh2 to enhance the expression of downstream-dependent genes, while this domain can also mediate protein–protein interactions [102].

MZF1 was first detected in humans by studies of hematopoietic cell development of the myeloid lineage. As the research continues to deepen, the development of other types of solid tumor cancer, such as colon cancer, is gradually recognized to be associated with MZMF1. Mudduluru et al. [74] first demonstrated that MZF1 overexpression in colorectal (RKO and SW480 cell lines) cancer cells can induce tumor cell proliferation, migration, and invasion by binding to the receptor tyrosine kinase (Axl) promoter gene, transactivating promoter activity and enhancing Axl mRNA and protein expression in a dose-dependent manner. A few years later, Deng et al. [103] concluded that, in colorectal (LoVo and SW480 cell lines) cancer cells, the transcription factor MZF1 potentiates their tumorigenic capacity by transcriptionally activating the p55PIK protein, a regulatory subunit of IA-PI3K, which, in turn, activates the PI3K/Akt pathway. Furthermore, a recent study confirmed that downregulated GSK3β in CRC cells can promote the expression of FTO by mediating FTO ubiquitination, FTO promotes the expression of the MZF1 protein by removing the m6A mRNA modification of MZF1, and the MZF1 protein activates the MZF1/C-MYC axis to advance cell proliferation and inhibit cell apoptosis [104]. Nevertheless, there are also relevant studies proposing that sulindac sulfide sensitizes cancer cells to trail-induced apoptosis by upregulating MZF1, inducing the expression of death receptor 5 (DR5) in tumor necrosis factor-related apoptosis inducing ligand (TRAIL), which interacts with adaptor proteins (e.g., FADD) and activates caspases to promote apoptosis in colon cancer cells [105]. In conclusion, MZF1 has both carcinogenic and tumor-suppressive effects on colon cancer. A deep understanding of its dual role can assist to develop a variety of therapies to target colon cancer. Additionally, this novel finding provides a cautionary note for the relevant drug development, especially the possible side effects of single-acting anticancer drugs.

## 6. Conclusions

As a large family of transcription factors, ZFPs are widely involved in various physiological and pathobiological processes in the human body. Colon cancer remains one of the most common malignant cancers worldwide, and there are still many unanswered questions in terms of early diagnosis and postoperative treatment. In recent decades, the properties of ZFPs to regulate transcription and control gene expression in colon cancer have been widely demonstrated, with more functional studies on ZFPs [18]. This article reviews the ZFPs structure, molecular mechanisms, functions, and the latest research progress in colon cancer. However, in the current state of the research, there are still many ZFPs whose mechanisms of action in colon cancer are not fully understood, and more mechanisms need to be further studied, such as ZNF146, ZNF511, ZNF346, ZNF638, ZNF700, and ZNF768 [39].

There are various types of zinc finger motifs, such as TAZ2 domain-like, Cys2His2 (C2H2)-like, zinc-binding loops, and Zn2/cys6, among which C2H2 zinc finger is the most widely distributed DNA-binding region. Within the C2H2 motif, ZFPs contain, in addition to zinc finger motifs, common domains such as the KRAB, set, and scan domains, which can function in binding to DNA, RNA, or proteins. Secondly, the expression of ZFPs can be upregulated or downregulated in cancer patients, which suggests that ZFPs may function as tumor suppressors or oncogenes simultaneously (Table 1). They can affect colon cancer development through a variety of biological processes, such as cell proliferation, EMT and liver metastasis, inflammation, the cell cycle, cancer stem cells, and DNA methylation. Moreover, in colon cancer, some ZFP functions seem to have a dual, opposing role to MZF1. Through multiple signaling pathways, MZF1 can both promote cancer multiplication and inhibit cancer progression through apoptosis (Figure 3). Therefore, due to the different roles of ZFPs in colon cancer, it is possible to invent specific inhibitors or agonists of ZFPs that interfere with the expression of their target genes, possibly providing new ideas for their research in colon cancer. In conclusion, ZFPs play an important role in the tumorigenic process of colon cancer, and their targeted agents need to be further explored.

## Figures and Tables

**Figure 1 cancers-14-05242-f001:**
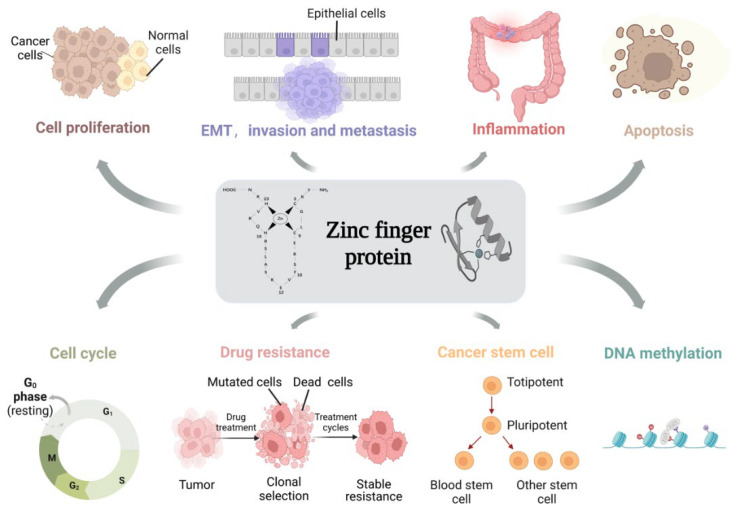
The function of ZFPs in regulating the cellular biological processes of colon cancer. ZFPs play important roles in the regulation of cell proliferation, epithelial–mesenchymal transition (EMT), invasion and metastasis, inflammation, cell cycle, cancer stem cells and DNA methylation in colon cancer cells. (This figure was created with biorender.com).

**Figure 2 cancers-14-05242-f002:**
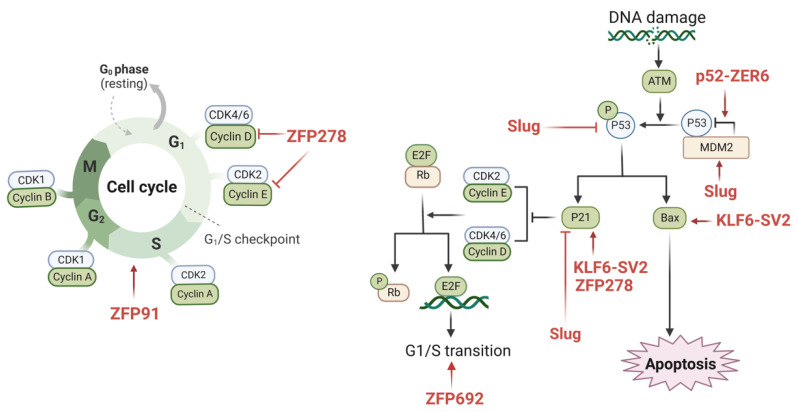
The possible mechanisms of ZFPs in regulating the cell cycle of colon cancer. There are various zinc finger proteins involved in cell cycle processes, such as ZFP91, ZFP278, ZFP692, Slug, KLF6-SV2, and P52-ZER6. The underlying mechanisms involve cyclin D, cyclin E, E2F, p21, p53, Bax, and MDM2. The underlying mechanism affects multiple molecules, such as cyclin D, cyclin E, E2F, p21, p53, Bax, and MDM2, eventually inducing cell progression or inhibiting cell proliferation in colon cancer. These ZFPs have great potential as novel therapeutic targets for colon cancer. (This figure was created with biorender.com).

**Figure 3 cancers-14-05242-f003:**
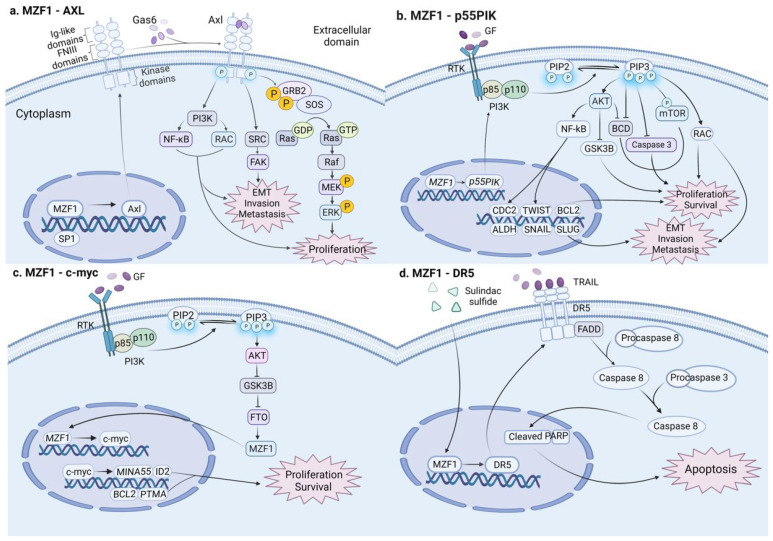
In colon cancer, MZF1 plays dual and opposite roles in different signaling pathways: (**a**) MZF1 transcriptionally activates the downstream target gene Axl and stimulates various signaling pathways, such as PI3K, FAK, Grb2/Ras, MEK/ERK, advancing cell proliferation, EMT transformation, invasion, and metastasis in colon cancer. (**b**) MZF1 transcriptionally activates the downstream target gene p55PIK; stimulates diverse signaling pathways such as PI3K/Akt and PI3K/RAC; and activates a series of downstream target genes such as CDC2, ALDH, BCL2, TWIST, SNAIL, and SLUG, promoting cell proliferation, EMT transformation, invasion, and metastasis in colon cancer. (**c**) MZF1 transcriptionally triggers the downstream target gene c-myc and multiple downstream target genes, such as MINA53, ID2, BCL2, and PTMA, promoting proliferation in colon cancer. (**d**) Sulfide sulindac sulfide induces the upregulation of MZF1. MZF1 promotes the expression of DR5 that interacts with FADD to activate caspases, promoting cell apoptosis and eventually inhibiting metastasis in colon cancer. (This figure was created with biorender.com).

**Table 1 cancers-14-05242-t001:** Representative ZFPs and the related signaling pathways in colon cancer.

ZFPs	Aliases	Expression	Biological Functions	Targets	References
ZNF143	pHZ-1, SBF, STAF	↓	EMT, invasion and metastasis	ZEB1, CDH1	[38]
ZNF146	OZF	↑	-	-	[39]
ZNF398	KIAA1339, P51, P71, p52-ZER6	↑	Tumorigenesis, cell proliferation and cell cycle	p53	[40]
KLF4	EZF, GKLF	↓	Cell proliferation	Bmi1	[41,42,43]
ZFP91	PZF, ZNF757, HPF7, HTF10	↑	Tumorigenesis, cell proliferation and inflammation	NF-κB/p65, HIF-1α, IL-1β	[44,45,46,47,48]
MAZ	Pur-1, ZF87, Zif87, ZNF801	↑	Inflammation	HIF-2a, Tnfa, Cxcl1, STAT3	[49]
ZNF185	SCELL	↑	Cell proliferation and liver metastasis	-	[36,37]
PATZ1	dJ400N23, MAZR, PATZ, RIAZ, ZSG ZBTB19, ZNF278,	↑	Potential proto-oncogene, cell proliferation and cell cycle	MAPK/ERK pathway	[50]
Twist	-	↑	Cancer stem cell and EMT, invasion and metastasis	Fibronectin, Vimentin, Snail, E-cadherin, N-cadherin,	[51,52]
Slug	-	↑	Cell cycle	Mdm2, P53, P21	[53]
ZEB1	AREB6, Zfhep, BZP, FECD6, NIL-2-A, ZEB, TCF8, Zfhx1a, PPCD3	↑	liver metastasis, DNA methylation, EMT, invasion and metastasis	CTBP, BRG1, uPA, PAI-1 E-cadherin, HDAC1, DNMT1	[30,32,36,38,54,55,56]
Snail	SNAI1	↑	-	E-cadherin, G9a, LSD1, HDACs, SUV39H1 inhibitor, PRC2	[57,58,59,60]
ZEB2	KIAA0569, SIP-1, SIP1, ZFHX1B	↑	EMT, invasion and metastasis	CTBP, E-cadherin	[54,60]
ZEB2-AS1		↑	Cell proliferation, apoptosis, EMT, invasion and metastasis	β-Catenin	[61]
CIZ1	LSFR1, ZNF356	↑	EMT, invasion and metastasis	-	[21]
ZFX	ZNF92	↑	Cell proliferation, cancer stem cell, EMT, invasion and metastasis	MAPK/ERK, PI3K/Akt, STAT3 pathway	[8,62,63]
RPS27	MPS-1, MPS1, S27	↑	EMT, invasion and metastasis	JNK/c-Jun signaling pathway	[33]
ZNF511	MGC30006, ZFP511	↑	-	-	[39]
ZNF217	ZABC1	↑	Oncogene, liver metastasis, EMT, invasion and metastasis	E-cadherin	[22]
ZNF70	Cos17, MGC48959	↑	Inflammation	NLRP3, IL-1β, inflammasome, STAT3	[64]
ZFP36	G0S24, NUP475, RNF162A, TIS11, TTP	↓	Cell proliferation, EMT, invasion and metastasis	IL-23, VDR, COX-2, VEGF, SOX9, MACC1, N-cadherin, ZEB1, Vimentin, E-cadherin, ZO-1	[30,31]
KLF12	AP-2rep, AP2REP, HSPC122	↑	Tumorigenesis and cell proliferation	-	[16]
KLF6-SV2	ZFP9	↓	Cell proliferation and cell cycle	Bax, p21	[18]
MZF1	MZF1A, MZF1B, ZFP98, ZSCAN6, ZNF42,	↑	Cell proliferation, apoptosis, EMT, invasion and metastasis	Axl, p55PIK, DR5, FADD Caspases	[17]
ZKSCAN3	ZF47, ZFP47, ZNF306, ZNF309, ZSCAN35	↑	EMT, invasion and metastasis	Integrin β4 and VEGF	[19]
ZNF692	AREBP, FLJ20531	↑	Cell proliferation, EMT, invasion and metastasis	CDK2, cyclin D1, Mmp-9, p27Kip1	[20]
ZNF750	FLJ13841	↑	Cell proliferation, cell apoptosis, EMT, invasion and metastasis	CYTOR	[65]
ZNF545	ZFP82, KIAA1948, MGC45380	↓	Apoptosis	KAP1	[66]
ZNF281	ZBP-99	↑	EMT, invasion and metastasis	IL-8, IL-1β, IL-17, IL-23,SNAIL, α-SMA Slug, TIMP-1, Vimentin, fibronectin, α-SMA	[4]

↓ indicates that the protein is down-regulated in colon cancer cell lines. ↑ indicates that the protein is up-regulated in colon cancer cell lines.

## Data Availability

All data included in this study are available upon request by contact with the corresponding author.

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
