# Peer review of "Zinc Finger Proteins: Functions and Mechanisms in Colon Cancer"

_cancers, 2022, doi:10.3390/cancers14215242_

Round 1
Reviewer 1 Report
Overall an interesting subject. That said the clarity could be improved- it gives a detaile overview but in most pragraphs it is more of enumeration than contexutalized read.
Do not understand why data from USa were used as a reference for CRC, since the article is a review and CRC has a global impact, also data on stages and survival seem to be a bit off- it is uncommon to consider stage II and II together (introduction).
To sum up, interesting subject a bit of work on clarity and clinical data should improve the manuscript considerably.
Author Response
Dear reviewer:
We thank you very much for giving us an opportunity to revise the manuscript titled “Zinc finger proteins: functions and mechanisms in colon cancer (ID: cancers-1977485)” and appreciate your reviews and comprehensive comments on our manuscript. Following the constructive comments item by item, we have made a substantial revision to the manuscript. By responding to the assistant editor and reviewers, we made appropriate revisions to the content of the article in response to the comments made. In the process of revision, we carefully checked whether all the references were relevant to the manuscript content and invited a colleague who is fluent in English writing to polish the article. By taking the advice from reviewer 1, We changed the global epidemiological data on colon cancer and the latest data on survival rates for colon cancer patients. As for the comments of reviewer 2, we unified the abbreviation of zinc finger protein (ZFPs) and added the full name of MZF1 in the whole paper. Additionally, the figure (Fig. 3) has been re-drawn with larger font size. We also have corrected some minor things and made additional changes that should improve the consistency and logicality of this manuscript.
All amendments are highlighted in red in the revised manuscript. After this comprehensive revision, we hope that the concerns of the reviewers have been answered properly, and the current manuscript version is scientifically more solid and hopefully meets the approval requirements of your journal.
Please see more details in our responses to the referee comments. We really appreciate your re-consideration of publishing this work in Cancers.
Sincerely yours,
Hongping Chen, M.D., PhD
Department of Histology and Embryology, Medical College, Nanchang University, 461 Bayi Road, Nanchang, Jiangxi, 330006, P.R. China
Telephone: +86-13517091601
E-mail: [email protected]
Below, we provide a point-by-point response to both reviewers’ comments.
Comments of Reviewer 1:
Question 1: Overall an interesting subject. That said the clarity could be improved- it gives a detailed overview but, in most paragraphs, it is more of enumeration than contextualized read.
Response: Thank you very much for your overall positive review of our work and for giving such detailed writing guidance! The correction has been made accordingly.
In accordance with this comment, we adjusted the text. We have added a lot of logical and summary content, specifically connectives, transitional sentences, and summary sentences.
(See tracked version P3 line149-150,160-161; P8 line277-278, 280-281,317-318; P9 line497,505-506)
Question 2: Do not understand why data from USA were used as a reference for CRC, since the article is a review and CRC has a global impact.
Response: Thank for pointing this out. In accordance with your suggestion, we used colon cancer data of GLOBOCAN 2018 database published by the International Center for Research on Cancer (IARC), which is global based. And we updated the references. (See tracked version P1 line41-44)
Question 3: Data on stages and survival seem to be a bit off- it is uncommon to consider stage II and III together (introduction).
Response: This is really a loose question, thank you for pointing it out. In accordance with your suggestion, we present the 5-year survival rates of colon cancer stage II and stage III separately. Previously, we put them together because they had similar 5-year survival rates, which may be due to the highly positive treatment attitude of stage III patients, the pertinence of treatment means, etc. However, it is not common to do so and we have made adjustments. (See tracked version P2 line64-68)
Question 4: To sum up, interesting subject a bit of work on clarity and clinical data should improve the manuscript considerably.
Response: We have revised the content of this article in a wide range according to your suggestion. In terms of clarity, we have added logical sentence segments to EMT, invasion and metastasis, inflammation, and apoptosis in the main text. Conjunctions and transitional sentences are added to almost all the text. For clinical data, we used IARC global colon cancer data and then referred to the five-year survival staging data for colon cancer patients from more authoritative publications.
Revising and self-editing:
We have noticed that your rating of “Is the English used correct and readable” is only 3 stars / 5 stars. It is speculated that the fluency and readability of our paper need to be further improved and we are supposed to make some modifications to the writing of the paper.
Response: We asked for help from one of the indicated graduate students majoring in English translation, who had received CATTI (China Aptitude Test for Translators and Interpreters) Level 3 certificate in English translation and had strong professional ability. After grinding it, the quality of the English language has improved considerably. (See tracked version 271-274,298-304)

Reviewer 2 Report
The manuscript authored by Liu et al. titled “Zinc finger proteins: functions and mechanisms in colon cancer”, authors reviewed the functions and mechanisms of Zinc finger proteins in colon cancer. The manuscript is well organized and written properly. The figures are designed conclusively. The following points need to be address.
1. Use only one abbreviation for zinc finger proteins. Authors are using ZNFs/ZFPs. Write full form of MZF1 in the abstract.
2. Font size in figure 3 is very small.
Author Response
Dear editors and reviewers:
We thank you very much for giving us an opportunity to revise the manuscript titled “Zinc finger proteins: functions and mechanisms in colon cancer (ID: cancers-1977485)” and appreciate your reviews and comprehensive comments on our manuscript. Following the constructive comments item by item, we have made a substantial revision to the manuscript. By responding to the assistant editor and reviewers, we made appropriate revisions to the content of the article in response to the comments made. In the process of revision, we carefully checked whether all the references were relevant to the manuscript content and invited a colleague who is fluent in English writing to polish the article. By taking the advice from reviewer 1, We changed the global epidemiological data on colon cancer and the latest data on survival rates for colon cancer patients. As for the comments of reviewer 2, we unified the abbreviation of zinc finger protein (ZFPs) and added the full name of MZF1 in the whole paper. Additionally, the figure (Fig. 3) has been re-drawn with larger font size. We also have corrected some minor things and made additional changes that should improve the consistency and logicality of this manuscript.
All amendments are highlighted in red in the revised manuscript. After this comprehensive revision, we hope that the concerns of the reviewers have been answered properly, and the current manuscript version is scientifically more solid and hopefully meets the approval requirements of your journal.
Please see more details in our responses to the referee comments. We really appreciate your re-consideration of publishing this work in Cancers.
Sincerely yours,
Hongping Chen, M.D., PhD
Department of Histology and Embryology, Medical College, Nanchang University, 461 Bayi Road, Nanchang, Jiangxi, 330006, P.R. China
Telephone: +86-13517091601
E-mail: [email protected]
Comments of Reviewer 2:
The manuscript authored by Liu et al. titled “Zinc finger proteins: functions and mechanisms in colon cancer”, authors reviewed the functions and mechanisms of Zinc finger proteins in colon cancer. The manuscript is well organized and written properly. The figures are designed conclusively. The following points need to be address.
Question 1: Use only one abbreviation for zinc finger proteins. Authors are using ZNFs/ZFPs. Write full form of MZF1 in the abstract.
Response: Thank you very much for giving our work a generally favorable evaluation,and it is so kind of you to provide us with such detailed improvements. Corrections have been made accordingly. As your suggestion, we have used ZFPs instead of ZNFs in text content (See revised version full text) and used full form of MZF1-Myeloid zinc finger 1 in the abstract (See tracked version P1 line30).
Question 2: Font size in figure 3 is very small.
Response: I'm sorry, this is really a problem that affects the reading experience. In accordance with your comments, we have enlarged the font size in the Figure 3.
(See tracked version P12 line628)
Revising and self-editing:
Thank you again for your high comments on the overall satisfaction with our review, however, we have noticed that your rating of “Is the English used correct and readable” is 4 stars / 5 stars. It is speculated that the fluency and readability of our paper still need to be further improved.
Response:For English writing, our paper was edited professionally by an indicated graduate student majoring in English translation, who had received CATTI (China Aptitude Test for Translators and Interpreters) Level 3 certificate in English translation and had strong professional ability. After grinding it, the quality of the English language has improved considerably. (See tracked version 271-274,298-304)
